# Multiple myeloma risk variant at 7p15.3 creates an IRF4-binding site and interferes with *CDCA7L* expression

Ni Li[1,2], David C. Johnson[2], Niels Weinhold[3,4], James B. Studd[1], Giulia Orlando[1], Fabio Mirabella[2], Jonathan S. Mitchell[1], Tobias Meissner[5], Martin Kaiser[2], Hartmut Goldschmidt[4,6], Kari Hemminki[7,8], Gareth J. Morgan[3] & Richard S. Houlston[1,2]

Genome-wide association studies have identified several risk loci for multiple myeloma (MM); however, the mechanisms by which they influence MM are unknown. Here by using genetic association data and functional characterization, we demonstrate that rs4487645 G > T, the most highly associated variant ($P = 5.30 \times 10^{-25}$), resides in an enhancer element 47 kb upstream of the transcription start site of c-Myc-interacting *CDCA7L*. The G-risk allele, associated with increased *CDCA7L* expression ($P = 1.95 \times 10^{-36}$), increases IRF4 binding and the enhancer interacts with the *CDCA7L* promoter. We show that suppression of *CDCA7L* limits MM proliferation through apoptosis, and increased *CDCA7L* expression is associated with adverse patient survival. These findings implicate IRF4-mediated *CDCA7L* expression in MM biology and indicate how germline variation might confer susceptibility to MM.

[1] Division of Genetics and Epidemiology, The Institute of Cancer Research, Surrey SM2 5NG, UK. [2] Division of Molecular Pathology, The Institute of Cancer Research, Surrey SM2 5NG, UK. [3] Myeloma Institute for Research and Therapy, University of Arkansas for Medical Sciences, Little Rock, Arkansas 72205, USA. [4] Department of Internal Medicine V, University of Heidelberg, 69117 Heidelberg, Germany. [5] Department of Molecular and Experimental Medicine, Avera Cancer Institute, La Jolla, California 92037, USA. [6] National Centre of Tumor Diseases, 69120 Heidelberg, Germany. [7] German Cancer Research Center, 69120 Heidelberg, Germany. [8] Center for Primary Health Care Research, Lund University, SE-205 02 Malmo, Sweden. Correspondence and requests for materials should be addressed to R.S.H. (email: richard.houlston@icr.ac.uk).

Genome-wide association studies (GWAS) have frequently identified statistically significant associations within noncoding regions of the genome; however, the causal DNA sequence and their functional effects have only been determined in a few instances[1].

GWAS have so far identified 17 risk loci for multiple myeloma (MM), with the association signal at 7p15.3 containing *CDCA7L* (cell division cycle-associated seven-like protein) being shown to be highly robust and not confined to a particular MM subtype[2–5]. *CDCA7L* represents an attractive candidate for the functional basis of the 7p15.3 association because of its role as a MYC-interacting protein, acting as a binding partner of LEDGF/p75 (lens epithelium-derived growth factor) and potentiating MYC-mediated transformation events[6–9]. Given that deregulation of *MYC* typifies plasma cell neoplasms, *CDCA7L* therefore represents an attractive functional basis of the 7p15.3 association.

Here we report the identification of the rs4487645 variant as the potential functional basis of the 7p15.3 association for MM. We demonstrate that through differential IRF4 binding rs4487645 influences expression of c-Myc-interacting *CDCA7L*. Increased *CDCA7L* expression was associated with increased cell proliferation and poorer patient survival. These findings provide insight into the mechanistic basis of germline susceptibility to MM.

## Results

**Fine mapping and epigenomic profiling of the risk locus.** We have analysed a large meta-analysis of six MM GWAS conducted in European populations totalling 9,866 cases and 239,188 controls[5]. Imputation of these GWAS including use of 1000 Genomes Project and UK10K data as reference[5] confirmed the previously identified single-nucleotide polymorphism (SNP) rs4487645 as providing the strongest association signal at the 7p15.3 risk locus[2] ($P = 5.30 \times 10^{-25}$, odds ratio = 1.24; logistic regression followed by fixed-effects meta-analysis, risk allele frequency = 0.65; Fig. 1). In addition, we have excluded the existence of multiple independent statistical signals at the 7p15.3 risk locus by performing association testing conditional on rs4487645 genotypes, observing no additional significant variants. Previously, we demonstrated an expression quantitative trait locus (eQTL) signal between rs4487645 and *CDCA7L* (ref. 10), with the risk allele associated with increased expression of *CDCA7L*. To further substantiate this eQTL, we analysed mRNA expression data on CD138-purified plasma cells from 1,449 MM cases from three independent series. Besides confirming that the association of rs4487645 with *CDCA7L* was the strongest 7p15.3 eQTL ($P = 1.95 \times 10^{-36}$; linear regression followed by fixed-effects meta-analysis; Supplementary Table 1 and Supplementary Fig. 1), conditional analysis provided no evidence for additional independent signals[10].

To further prioritize candidate MM risk variants, we examined the regulatory potential of SNPs in linkage disequilibrium ($r^2 \geq 0.8$) with rs4487645. We prioritized credible candidate risk variants by their regulatory potential inferred from B-cell-specific DNase I hypersensitivity (HS), formaldehyde-assisted isolation of regulatory elements (FAIRE), promoter-/enhancer-associated histone marks and chromatin immunoprecipitation sequencing (ChIP-seq; Fig. 1 and Supplementary Table 2)[11]. SNP rs4487645 resides within a 1 kb active enhancer region as predicted by ChromHMM, supported by open chromatin analysis (DNase HS and FAIRE), as well as H3K4Me1, H3K4Me3 and H3K27Ac peaks. ENCODE ChIP-seq data confirmed that rs4487645 localizes within an IRF4/PU. 1-binding motif ETS/ISRE (ETS/interferon-stimulated-response-element composite DNA motif)-consensus element (EICE; 5′-G[G/T]AANNGAAA-3′).

No other candidate MM risk variant at the 7p15.3 locus showed the unique combination of highly significant GWAS and eQTL signal, open chromatin, active enhancer localization and transcription factor binding. By analysing the germline exomes of 463 MM cases, we only identified one CDCA7L mutation (S197Y) in one sequenced individual[12], thus excluding the possibility that the 7p15.3 association signal is a consequence of linkage disequilibrium with a rare disease-causing coding variant. Collectively, these data suggest that rs4487645 is the single best causal SNP candidate for the 7p15.3 association.

**SNP rs4487645 modulates IRF4-binding affinity.** Using Find Individual Motif Occurrences, we assessed the genomic sequence overlapping rs4487645 with the DNA-binding position weight matrices of multiple human transcription factors[13]. In this analysis, the IRF4 position weight matrix was the strongest match for the sequence overlapping rs4487645, with a preference for the risk-associated G-allele (G-allele Q value = $9.39 \times 10^{-5}$; T-allele Q value = 0.017; Fig. 2a). To examine for differential IRF4 binding between the G and T alleles of rs4487645 and its binding motif, we performed ChIP–qPCR (quantitative PCR) in the GM11992 lymphoblastoid cell line, which is heterozygous for rs4487645. The G-allele of rs4487645 showed significant preferential binding to IRF4 (two-tailed *t*-test, $P = 0.0014$; Fig. 2b), supporting a role for rs4487645 in modulating IRF4-binding affinity on the site of overlap.

**SNP rs4487645 is localized within an enhancer element.** We next performed luciferase reporter assays to determine the effect of rs4487645 on enhancer activity in the MM cell line KMS11. Transfection of constructs containing the risk G-allele showed significant enhancement of normalized luminescence compared with the T-allele (two-tailed *t*-test, $P = 0.031$; Fig. 2c,d), consistent with a model in which the causal risk variant is associated with increased *CDC7AL* expression.

**rs4487645 shows long-range interaction with *CDCA7L* promoter.** Physical interactions between regulatory elements and promoters have a major role in regulating gene expression[14,15]. Following on from our observation that rs4487645 is within an IRF4-bound enhancer element, and that rs4487645 and the *CDCA7L* promoter retain active regulatory chromatin marks (Fig. 2e)[16], we interrogated using 3C-qPCR whether rs4487645 physically interacts with the *CDCA7L* promoter in GM11992 and KMS11. The genomic region to which rs4487645 maps was demonstrated to form a chromatin-looping interaction with the *CDCA7L* promoter in both cell lines (Fig. 3a,b), suggesting a mechanism by which IRF4-bound rs4487645 regulates *CDCA7L* expression.

***IRF4* regulates *CDCA7L* expression.** *CDCA7L* expression is significantly correlated with *IRF4* expression in CD138-purified MM cells (Mann–Whitney–Wilcoxon test, $P = 0.0012$; Supplementary Fig. 2). To establish a direct relationship between *IRF4* and *CDCA7L* expression, we performed small interfering RNA (siRNA) experiments in KMS11. Knockdown of *IRF4* was accompanied by a significant reduction in *CDCA7L* mRNA (two-tailed *t*-test, $P = 0.0038$; Fig. 3c and Supplementary Fig. 3). Collectively, these data are consistent with *CDCA7L* being a downstream target of *IRF4* in MM, with increased binding of IRF4 at the G-risk allele of rs4487645 leading to increased *CDCA7L* expression.

***CDCA7L* affects myeloma proliferation and patient survival.** SNP rs4487645 and the *CDCA7L* promoter localize to H3K27Ac

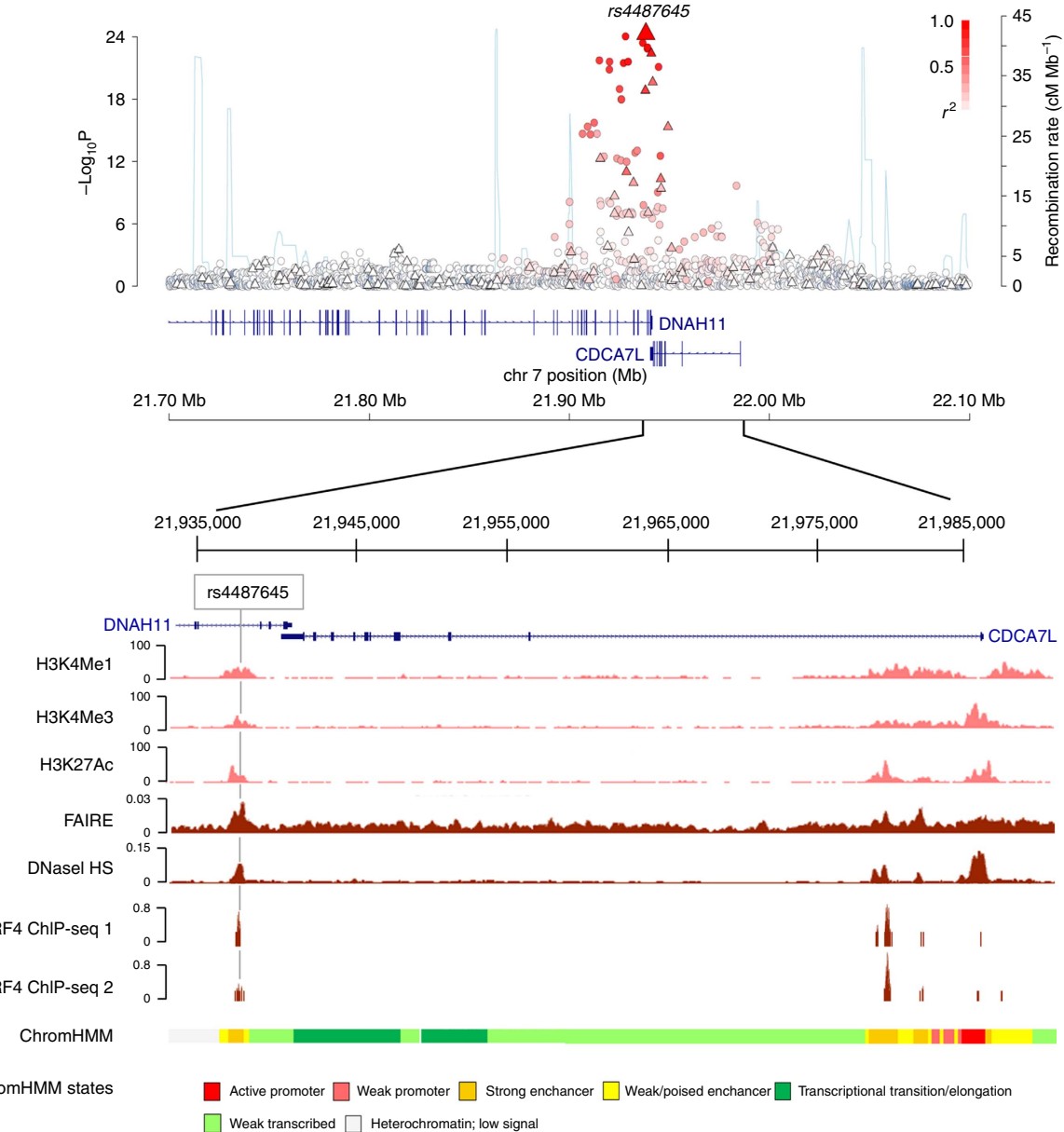

**Figure 1 | Genetic mapping and epigenetic landscape at the 7p15.3 locus.** Manhattan plot of 7p15.3 is shown with genotyped (triangles) and imputed (circles) SNPs. $-\log_{10} P$ values (y axis) of the SNPs are shown according to their chromosomal positions (x axis; NCBI build 37 of the human genome). The top genotyped SNP (rs4487645) is labelled with a large triangle. The colour intensity of each symbol in the Manhattan plot reflects the extent of linkage disequilibrium with rs4487645, from white ($r^2 = 0$) to dark red ($r^2 = 1.0$). Genetic recombination rates are estimated using HapMap Utah residents of Western and Northern European ancestry (CEU) samples, and shown with a light blue line. The relative positions of *DNAH11* and *CDCA7L* transcripts mapping to 7p15.3 are shown. The chromatin state of the *DNAH11–CDCA7L* gene locus in GM12878 is detailed with H3K4Me1, H3K4Me3, H3K27Ac, FAIRE, DNaseI HS studies and chromatin state segmentation track (ChromHMM), with called peaks on two replicates of IRF4 ChIP-seq in GM12878 shown below accessed from the UCSC Genome Browser. Position of rs4487645 is highlighted with grey line.

and H3K4Me3 chromatin marks and are enriched for MYC and RNA polymerase II binding in MM1.S cells (Fig. 2e). Since high levels of *MYC* were associated with high levels of *CDCA7L* in MM (Mann–Whitney–Wilcoxon test, $P = 1.32 \times 10^{-4}$; Supplementary Fig. 4), this implies a potential network involving *MYC*, *IRF4* and *CDCA7L* at the 7p15.3 association region. A significant correlation between *CDCA7L* expression and cellular proliferation was shown as evidenced by the GPI in German ($n = 658$) and US ($n = 608$) MM data sets (Pearson's product–moment correlation, $P < 1.0 \times 10^{-16}$; Supplementary Fig. 5). To directly investigate the effect of *CDCA7L* expression on malignant phenotype, we knocked down *CDCA7L* levels in

KMS11 using short hairpin RNA (shRNA; Supplementary Fig. 6). Reduction of *CDCA7L* expression was associated with reduced cell proliferation (two-tailed t-test, $P < 0.05$; Fig. 4a,b) and increased apoptosis (two-tailed t-test, $P < 0.05$; Fig. 4c,d). We observed a 39% and 71% increase of apoptotic cells in the shRNA-CDCA7L-1 and shRNA-CDCA7L-2 knockdown cell lines, respectively, relative to control. In contrast, there was no difference between the proportions of cells at different stages of the cell cycle between *CDCA7L* knockdown and control cell lines (Supplementary Fig. 7). Following on from this we examined the impact of *CDCA7L* expression on MM patient outcome in five independent series totalling 1,573 cases. High levels of *CDCA7L*

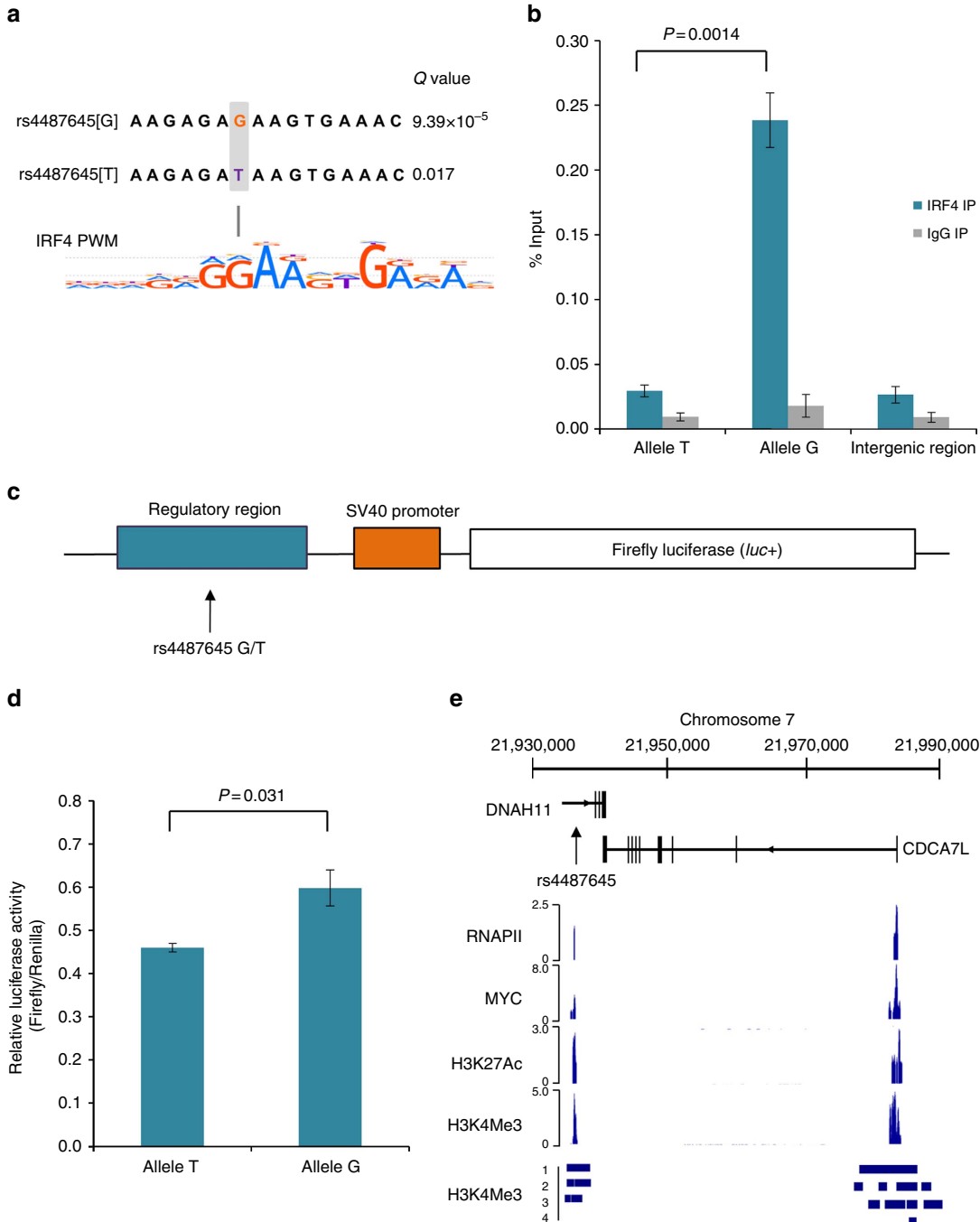

**Figure 2 | SNP rs4487645 shows allele-specific binding of IRF4 and confers enhancer regulatory activity in multiple myeloma cells.** (**a**) rs4487645 is located in an IRF4 DNA-binding motif predicted by Find Individual Motif Occurrences (FIMO). Shown below is the IRF4 position weight matrix generated by HOCOMOCO. G-risk allele is highlighted in red. (**b**) ChIP–qPCR using either the anti-IRF4 or isotype control antibody was performed on GM11992 heterozygous for rs4487645. The ChIP–qPCR signal is based on the abundance of allele-specific amplicons for rs4487645 in the immunoprecipitated DNA, relative to input DNA. Also shown is the relative abundance of intergenic control region. ChIP data show differential binding of IRF4 for the risk G-allele relative to the T-allele. Data shown are mean ± s.e.m. from three biological replicates and assessed by two-tailed *t*-test. Each qPCR reaction was performed with three technical replicates. (**c**) An ∼1 kb putative regulatory sequence flanking rs4487645 (G/T) was cloned upstream of the SV40 promoter in the pGL3-promoter vector for testing luciferase reporter gene activity. (**d**) The resultant reporter constructs were transiently transfected into KMS11, and the relative luciferase activity was measured for each reporter gene construct. The luminescence ratio of the experimental vector to the Renilla internal control, pRL-SV40, was normalized to the backbone pGL3-SV40 promoter vector. Data shown are mean ± s.e.m. from four biological replicates and assessed by two-tailed *t*-test. (**e**) The chromatin state of the *DNAH11–CDCA7L* gene locus is detailed with RNA polymerase II (RNAPII), MYC, H3K4Me3 and H3K27Ac ChIP-seq data from MM cell line MM1.S (GSE36354). ChIP-seq data in four MM patients (labelled 1–4) show MM-unique H3K4Me3-occupied regions compared with normal bone marrow plasma cells (GSE53215).

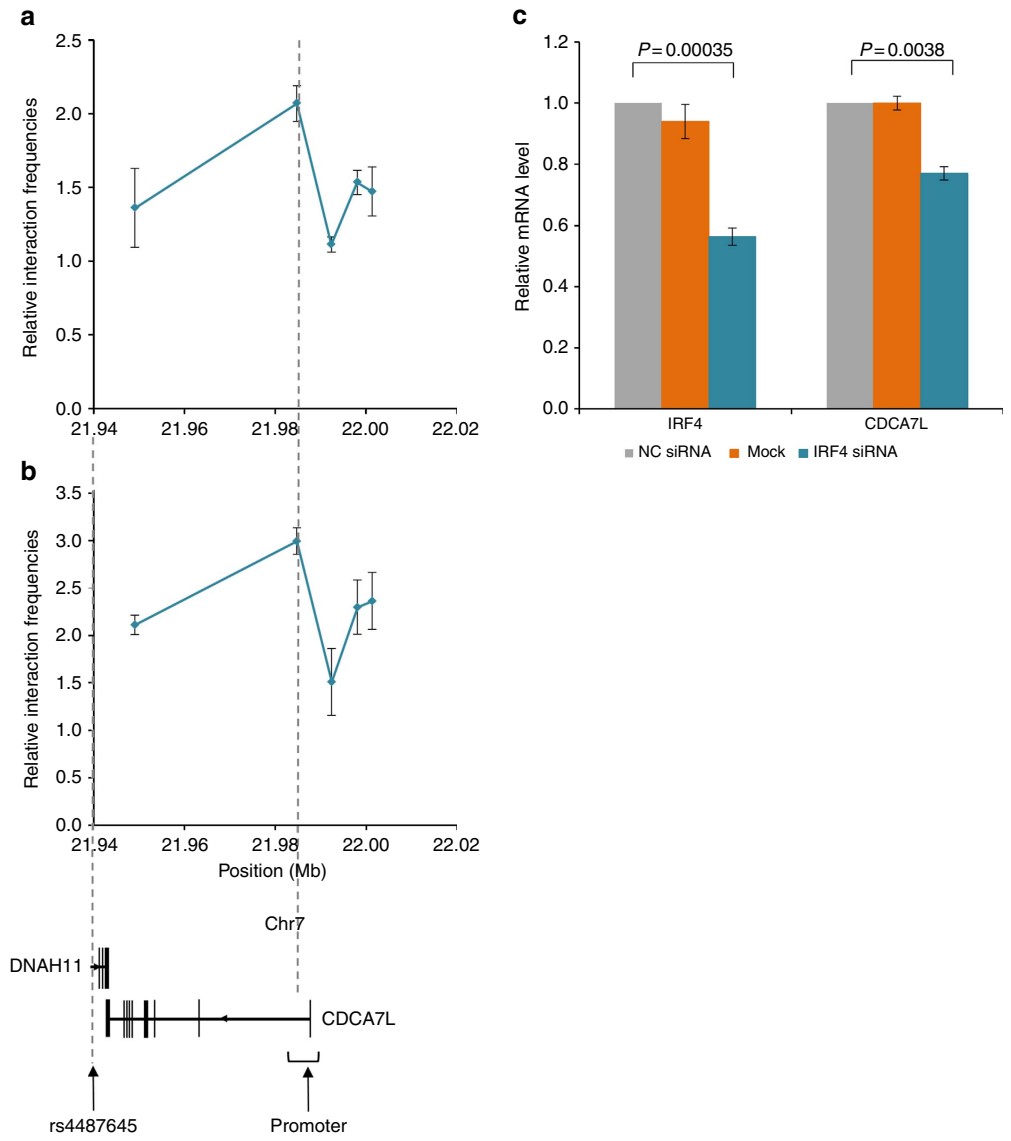

**Figure 3 | Physical interaction between SNP rs4487645 and *CDCA7L* promoter and the regulation of *IRF4* on *CDCA7L*.** Long-range interactions detected between fragments mapping to rs4487645 and the *CDCA7L* promoter in GM11992 (**a**) and KMS11 (**b**). Relative interaction frequencies between rs4487645 and target regions ∼70 kb upstream were determined by 3C-qPCR, comparing the relative abundance of ligation products formed between the fragment mapping to rs4487645 and each of the target fragments ± s.e.m., normalized to the relative abundance of intersite control region. Each qPCR reaction was performed with three technical replicates. The assay was performed independently for three times in both GM11992 and KMS11. (**c**) qPCR analysis of the siRNA knockdown of *IRF4*. Data shown are the mean *IRF4* and *CDCA7L* mRNA levels ± s.e.m. relative to the *GAPDH* reference mRNA level, normalized to control siRNA (NC siRNA). Data also show relative mRNA level of *IRF4* and *CDCA7L* for mock transfection without siRNA oligos. Each qPCR reaction was performed with three technical replicates. *P* values were determined with two-tailed *t*-test over three biological replicates.

expression were associated with poorer overall survival in MM patients (Cox regression, $P = 3.55 \times 10^{-4}$, hazard ratio = 1.52, 95% confidence interval: 1.21–1.90; Supplementary Fig. 8).

## Discussion

Collectively, our data demonstrate that the underlying molecular mechanism for the 7p15.3 MM risk locus is mediated through rs4487645, which resides within a transcriptional enhancer and disrupts an IRF4 transcription factor-binding site. Our data are compatible with the rs4487645 G-allele, conferring an increased MM risk through increased IRF4-mediated expression of *CDCA7L*. Furthermore, epigenetic and 3C data are consistent with rs4487645 localizing within a chromatin contact domain and forming a 'loop domain', which is expected to bring the

region into physical contact with the promoter of *CDCA7L*. Mechanistically, our data also provide linkage between the G-risk allele, which is associated with higher expression of *CDCA7L* in MM, with effects on cell proliferation and response to apoptosis. In patients the increased tumour proliferation is reflected in a clinical phenotype characterized by adverse survival.

Importantly, our study highlights a previously unknown functional interaction between *CDCA7L* and *IRF4* in MM. *IRF4* is a key transcriptional regulator in B-cell development with critical functions in plasma cells. Moreover, *IRF4* is an essential MM gene directing a broad expression programme; knockdown of *IRF4* induces rapid non-apoptotic cell death[17]. *IRF4* and *MYC* also form a positive regulatory loop in MM, reinforcing expression of each other. We demonstrate a correlation between *MYC* and *CDCA7L* expression in MM patients, and

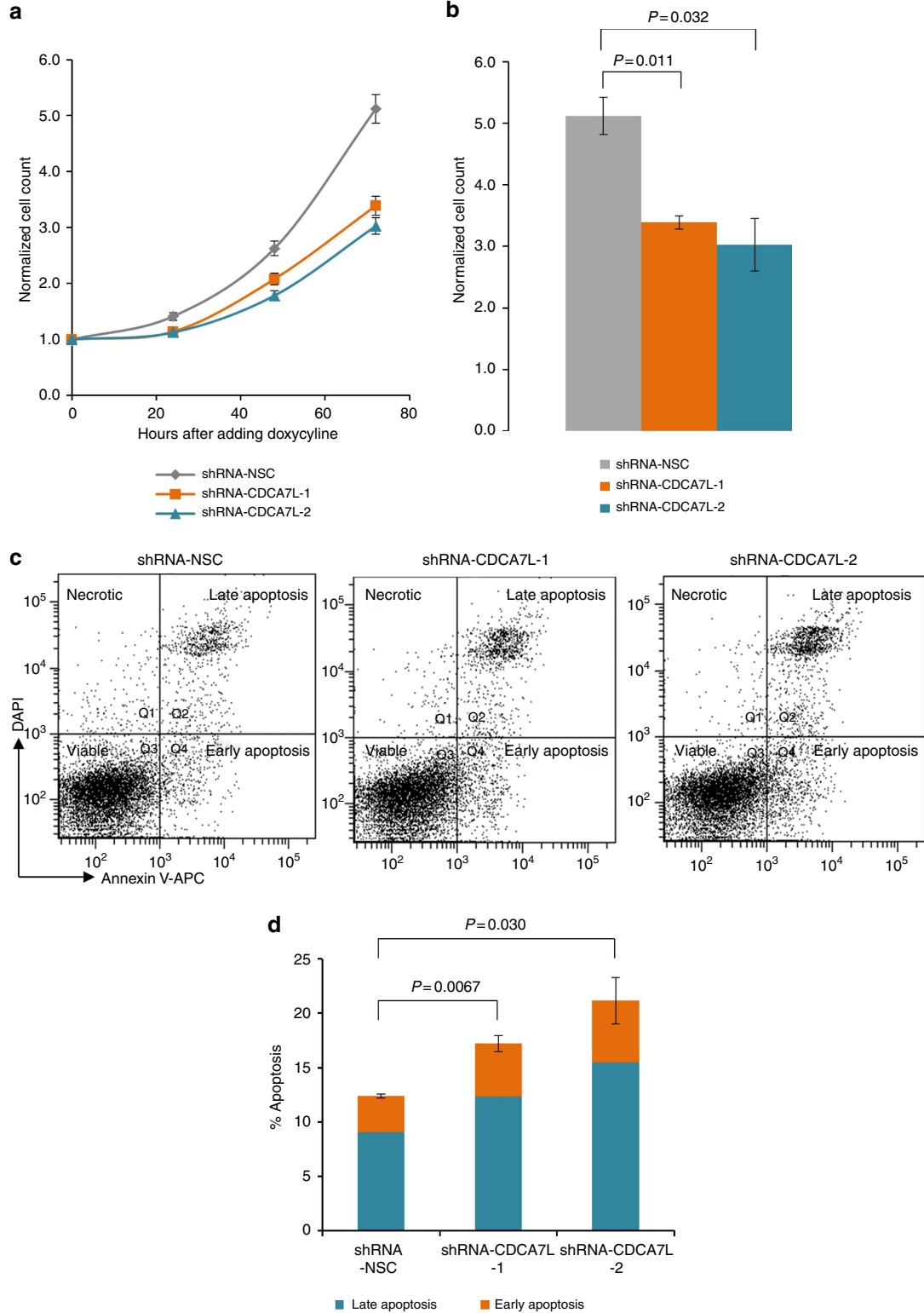

**Figure 4 | *CDCA7L* knockdown induces cell apoptosis and suppresses cellular proliferation.** (**a**) Data shown are the mean cell counts of viable Trypan-blue-negative cells ± s.e.m. for three biological replicates at 24, 48 and 72 h after addition of doxycycline (final concentration 1 μg ml$^{-1}$). Data were normalized to initial seeding number at 0 h when doxycycline was added to cells. (**b**) *P* values assessing the differences in normalized cell counts between *CDCA7L* and control knockdowns at 72 h were determined using two-tailed t-test. (**c**) Cell viability at 72 h after addition of doxycycline was assessed with FACS using Annexin V-APC and 4,6-diamidino-2-phenylindole. Representative fluorescence-activated cell sorting data show four subgroups of cells: the lower left quadrant (Q3; unstained) represents the viable cell population; the lower right (Q4; Annexin V-APC + DAPI − ) contains early apoptotic cells; the upper right quadrant (Q2; Annexin V-APC + DAPI + ) represents late apoptotic cells; the upper left (Q1; Annexin V-APC − DAPI + ) represents necrotic cells. (**d**) The percentages of apoptotic cells of shRNA-CDCA7L-1, shRNA-CDCA7L-2 and shRNA-NSC are shown at 72 h after addition of doxycycline. Data shown are mean ± s.e.m. for three biological replicates. *P* values were determined with two-tailed *t*-test.

show that MYC binds to the *CDCA7L* promoter. *CDCA7L* has been shown to be oncogenic in several cancers, in which MYC physically interacts and induces expression of *CDCA7L* (refs 8,9). Moreover, *CDCA7L* potentiates MYC-transforming activity, and can complement a transformation-defective MYC mutant[6].

In conclusion, we provide compelling evidence of the causal molecular mechanism for the 7p15.3 association for MM, implicating a more extensive pathway linking *CDCA7L*, *IRF4* and *MYC* in the development of this malignancy. The importance of enhancer-mediated *CDCA7L* expression in MM demonstrates the potential of translating basic mechanistic insights of tumour initiation towards development of therapeutic strategies. Indeed, downregulation of *IRF4* in MM has been shown to be central to potent antitumour activities by immunomodulatory drugs[18,19]. Similarly, bromodomain inhibition of the transcriptional co-activators CBP/EP300 is seen as a therapeutic strategy to target the *IRF4* network in MM[20]. Conclusively, the molecular interactions behind the 7p15.3 association potentially represent new targetable components of the MM oncogenesis pathway.

## Methods

**Ethics.** Collection of patient samples and associated clinicopathological information was undertaken with written informed consent and relevant ethical review board approval at respective study centres in accordance with the tenets of the Declaration of Helsinki. Specifically, the MRC Leukaemia Data Monitoring and Ethics committee (MREC 02/8/95, ISRCTN68454111, MREC 17/09/09, ISRCTN49407852), University of Heidelberg Ethical Commission (229/2003, S-337/2009, AFmu-119/2010) and the University of Arkansas for Medical Sciences Institutional Review Board (IRB 202077).

**ENCODE and chromatin state dynamics.** To explore the epigenetic profile of association signals, we used DNAse HS, FAIRE, transcription factor ChIP-seq data, histone modifications (H3K4Me1, H3K4Me3 and H3K27Ac) and chromatin state segmentation (ChromHMM) from the ENCODE project in GM12878 (ref. 11). We used HaploReg[21] and RegulomeDB[22] to examine whether any of the SNPs or their proxies (that is, $r^2 \geq 0.8$ in the 1000 Genomes EUR reference panel) annotates putative transcription factor binding or enhancer elements. We assessed sequence conservation using GERP[23] and PhastCons[24]. We accessed RNA polymerase II, MYC, H3K27Ac and H3K4Me3 ChIP-seq data performed on MM1.S from GSE36354 data set, and MM-unique H3K4Me3 ChIP-seq data in four MM patients compared with normal bone marrow plasma cells from GSE53215 data set.

**Expression quantitative trait loci analysis.** eQTL analyses were performed for genes within 1 MB of rs4487645 for CD138-purified plasma cells from 183 MRC Myeloma IX trial patients (GSE21349), 658 Heidelberg patients (E-MTAB-2299) and 608 US patients (GSE2658, GSE31161) using Affymetrix Human Genome U133 2.0 Plus Array[10] (Supplementary Table 3). Briefly, German, UK and US data were pre-processed separately, followed by analysis using a Bayesian approach to probabilistic estimation of expression residuals to infer broad variance components, accounting for hidden determinants influencing global expression (for example, copy number, translocation status and batch effects). The association between genotype of the sentinel SNP and gene expression of genes within 500 kb either side of rs4487645 was evaluated based on the significance of linear regression coefficients. We pooled data from each of the three studies under a fixed-effects model, controlling for false discovery rate (FDR) and calling significant associations with FDR ≤ 0.05. Conditional analysis was performed as described previously[10].

**GPI.** To investigate the relationship between *CDCA7L* expression and proliferation of MM plasma cells, we calculated GPI as recently described[25] using GC-RMA (GeneChip Robust Multiarray Averaging)-normalized German ($n = 658$) and US ($n = 608$) expression data.

**Relationship between gene expression.** We assessed the relationship between *IRF4* and *CDCA7L* gene expression (log₂-transformed) in five independent MM data sets GSE9782 ($n = 265$), GSE2658 ($n = 559$), GSE19784 ($n = 320$), E-MTAB-372 ($n = 246$) and GSE21349 ($n = 183$). The *CDCA7L* expression in patients within upper and lower quartiles of *IRF4* expression was compared using Mann–Whitney–Wilcoxon test for each data set. To assess the relationship between *MYC* and *CDCA7L*, *CDCA7L* expression in patients from the top and bottom quartiles of *MYC* expression in the five independent data sets was analysed by Mann–Whitney–Wilcoxon test. *P* values from the five patient data sets were combined by Fisher's Method. Statistical tests were conducted using the R software version 3.1.3.

**Cell lines.** Human MM cell line KMS11 was obtained from the American Type Culture Collection (ATCC) and grown in Advanced RPMI 1640 Medium (RPMI) supplemented with GlutaMAX and 10% heat-inactivated fetal bovine serum (FBS). Human lymphoblastoid cell lines GM11992 and GM12878 were obtained from Coriell Institute and grown in RPMI 1640 culture medium supplemented with 15% FBS. HEK293-T17 were obtained from ATCC and grown in DMEM supplemented with 10% heat-inactivated FBS. Cell culture media were obtained from Life Technologies (Carlsbad, CA, USA) and FBS from PAA Laboratories (Pasching, Austria). Cells were cultured at 37 °C, 100% relative humidity and 5% CO₂. Cell lines used were confirmed to be mycoplasma-free (PCR Mycoplasma Test Kit I/C, PromoCell, Heidelberg, Germany). Genotype of rs4487645 in KMS11, GM11992 and GM12878 was confirmed by Sanger sequencing using sequencing primers detailed in Supplementary Table 4.

**ChIP–qPCR assay.** Chromatin from GM11992 was immunoprecipitated with the ChIP-IT Express Enzymatic Kit (Active Motif, Carlsbad, CA, USA), using 3 μg anti-IRF4 (sc-6059X; Santa Cruz Biotechnology, TX, USA) and 3 μg normal goat IgG antibodies (sc-2028; Santa Cruz Biotechnology) per immunoprecipitation (IP) reaction. Immunoprecipitated chromatin was subjected to real-time qPCR using SYBR Green PCR Master Mix (Life Technologies). qPCR was conducted as follows: 50 °C 2 min, 95 °C 10 min, 40 cycles of 95 °C 15 s, 64 °C 1 min, end cycle 95 °C 15 s, 60 °C 15 s and 95 °C 15 s. Each IP was analysed with primers for rs4487645 (G/T) and intergenic control detailed in Supplementary Table 4. Primer efficiencies were corrected with a standard curve from serial dilutions of GM11992 genomic DNA. Data from IRF4 IP and control IP were presented as enrichment relative to input DNA. Each qPCR reaction was performed in three technical replicates, and the statistical significance was calculated over three biological replicates using two-tailed *t*-test.

**Plasmid construction and luciferase assays.** An ∼1 kb regulatory region with the non-risk allele of rs4487645 was amplified from human genomic DNA using primers detailed in Supplementary Table 4. Gel-purified PCR products (Qiagen, Hilden, Germany) were A-tailed using 2 U Thermoprime DNA polymerase (ThermoFisher Scientific, Waltham, USA) and 200 μM dATP for 30 min at 70 °C. The product was cloned into the PCR8/GW/TOPO vector and transformed into *Escherichia coli*. Single bacterial colonies containing the vector were cultured and purified (Qiagen Mini-prep Kit). The risk allele of rs4487645 was generated with site-directed mutagenesis using Quick Change XL (Agilent, Santa Clara, USA). Regulatory regions with both non-risk and risk alleles were cloned into pGL3 *luc2*-promoter vector using Gateway LR Clonase II (ThermoFisher Scientific). pGL3 *luc2* constructs were amplified in *E. coli* followed by purification of plasmid DNA using the QIAfilter Plasmid Midi Kit (Qiagen). Inserts and site-directed mutagenesis changes were Sanger-sequenced using primers detailed in Supplementary Table 4. The reporter constructs were introduced into KMS11 using nucleofection. In all, $5 \times 10^6$ cells were resuspended in 100 μl Cell Line Nucleofector Solution V with 3 μg reporter constructs and 60 ng of internal control plasmid (pRL-SV40). Nucleofections were performed using Amaxa Nucleofector I (X-01 programme; Lonza, Basel, Switzerland). Transiently transfected cells were cultured for 24 h, following which the luciferase assay was performed using the Dual-Luciferase Reporter Assay System (Promega, Madison, USA) as per the manufacturer's recommendations. Firefly and Renilla luciferase luminescence were measured sequentially on a Fluoroskan Ascent FL plate reader (ThermoFisher Scientific). The ratio of luminescence from the experimental reporter to the luminescence from the control reporter was calculated for each sample, defined as the relative luciferase activity. Statistical significance was calculated using two-tailed *t*-test over three biological replicates.

**In situ chromosome conformation capture.** 3C library preparation was performed as described in ref. 14 but with the following modifications. In all, $25 \times 10^6$ crosslinked GM11992 were resuspended in 1 ml of the Hi-C buffer for 45 min, followed by douncing with a 2 ml dounce homogenizer. Altogether, $25 \times 10^6$ crosslinked KMS11 cells were resuspended in 13 ml of the 4C lysis buffer[26] and incubated for 30 min on ice. DNA was digested overnight with rotation with 2,000 U Csp6I restriction enzyme (ThermoFisher Scientific). Restriction fragment overhangs were not filled or marked with biotin as described in ref. 14. Digestion reaction was aliquoted into 1.2 ml ligation reactions by combining 250 μl digestion mix with 950 μl ligation master mix in each. Each ligation reaction contained 6,000 U T4 DNA ligase (New England Biolabs, Ipswich, USA) for GM11992 or 10,000 U T4 DNA ligase for KMS11. Ligation was performed at 16 °C overnight at 900 r.p.m. After reverse crosslinking, DNA was extracted with phenol:chloroform:isoamyl alcohol (25:24:1) and precipitated in ethanol. RNAse (Roche, Basel, Switzerland) treatment was performed subsequently. Library quantity was assessed with the Qubit dsDNA BR Assay Kit (Life Technologies). Control library was made from RP11-163E10 BAC clone (Source Bioscience, Cambridge, UK). *E. coli* containing the BAC clone was re-streaked, picked and cultured. Following Maxiprep (Qiagen) purification, the BAC clone was further purified with phenol:chloroform:isoamyl alcohol (25:24:1) and ethanol precipitation. BAC clone (10 μg) was digested with 250 U Csp6I in 500 μl reaction, followed by ethanol precipitation. Digested DNA (20 μg) was randomly ligated with 7,600 U T4 DNA ligase (16 °C at 900 r.p.m. overnight), with

additional ligase and ATP next day (16 °C at 900 r.p.m.) as per the protocol with modifications[27]. Finally, the ligated BAC library was purified with phenol:chloroform:isoamyl alcohol.

Chromatin-looping was interrogated between fragments mapping to rs4487645 and the *CDCA7L* promoter generated from Csp6I digestion using SYBR Green PCR Master Mix (Life Technologies). Forward primers were designed over the ligation junction of a ligation product, with a constant reverse primer from the fragment mapping to rs4487645. Primer sequences are given in Supplementary Table 4. qPCR was conducted as follows: 95 °C 10 min, 40 cycles of 95 °C 30 s, 68 °C 30 s, end cycle 95 °C 15 s, 60 °C 15 s, 95 °C 15 s. Primer normalization was done using the randomly ligated BAC control template. Primers were designed for a region between two Csp6I cut sites (intersite control) to control for loading of each 3C experimental library. Each qPCR reaction was performed with three technical replicates, and the data presented were the average of three biological replicates for each cell line.

**siRNA knockdown.** siRNA targeting *IRF4* and control siRNA (NC siRNA) were obtained from Eurofins Genomics (Wolverhampton, UK) and detailed in Supplementary Table 4. KMS11 cells were transfected with either 200 nM siRNA or no siRNA (Mock transfections) using nucleofection as described in luciferase assays. Total RNA was extracted 24 h post-transfection using the RNeasy Plus Mini Kit (Qiagen) and reverse-transcribed using the SuperScript First Strand Synthesis Kit (Life Technologies) according to the manufacturer's recommendations. Knockdown efficiency was assessed using qPCR (1:20 dilution of cDNA as template) and western blotting using the standard protocols. Antibodies against IRF4, CDCA7L and β-actin for western blots are described below. Transcript levels of *IRF4* and *CDCA7L* expression were quantified using SYBR Green PCR Master Mix (Life Technologies) and normalized to *GAPDH*. A standard curve of serial dilutions of KMS11 cDNA was used to normalize for primer efficiencies. qPCR was conducted as follows: 50 °C 2 min, 95 °C 10 min, 40 cycles of 95 °C 15 s, 60 °C 1 min, end cycle 95 °C 15 s, 60 °C 15 s and 95 °C 15 s. Each qPCR reaction was performed with three technical replicates. Statistical significance was calculated using two-tailed *t*-test over three biological replicates. Primer sequences are detailed in Supplementary Table 4.

**Lentiviral transduction of shRNAs.** Lentivirus constructs were produced by transient transfection of HEK293-T17 cells as described previously[28,29]. Briefly, HEK293-T17 was co-transfected with the pTRIPZ-based shRNA vectors, pCMVR8.47 and pMD2.G by calcium phosphate precipitation. pTRIPZ-based doxycycline-inducible lentiviral vectors, V2THS_26553 (shRNA-CDCA7L-1 in Fig. 4) and V2THS_26555 (shRNA-CDCA7L-2) for *CDCA7L* knockdown, and non-silencing negative control RHS4743 (shRNA-NSC) were obtained from GE Healthcare Dharmacon (Little Chalfont, UK). The lentivirus particles were harvested, concentrated with Lenti-X Concentrator (Clonetech, Mountain View, USA) and used to infect KMS11. Transduced KMS11 were selected with 1 μg ml$^{-1}$ puromycin. The shRNA was induced from a tetracycline-inducible promoter with 1 μg ml$^{-1}$ doxycycline (Sigma, St Louis, USA). Expression of *CDCA7L* was assessed using qPCR and western blotting every 24 h post induction with doxycycline, with the optimal knockdown efficiency observed at 72 h.

**Western blotting.** Transduced KMS11 from *IRF4* or *CDCA7L* knockdown were collected and lysed in RIPA buffer (1% w/v sodium deoxycholate, 1% v/v Triton X-100, 1% v/v Nonident P-40, 0.1% SDS, 150 mM NaCl, 50 mM Tris pH 8.0 and 30 mM NaF) supplemented with Complete Protease inhibitor cocktail (Roche). Protein concentrations were determined with the BCA Protein Assay Kit (ThermoFisher Scientific). Ten micrograms of each protein lysate were loaded on NuPAGE Novex 10% Bis-Tris Protein Gels (ThermoFisher Scientific) and run in 1 × NuPAGE MOPS SDS Buffer (ThermoFisher Scientific). Proteins were transferred to polyvinylidene difluoride membranes in iBlot2 Transfer Stacks (ThermoFisher Scientific) using the P0 programme of the iBlot2 Blotting System (ThermoFisher Scientific) according to the manufacturer's recommendations. Western blotting was performed with anti-IRF4 (1:500 in 5% milk; sc-6059; Santa Cruz Biotechnology), anti-CDCA7L (1:2,000 in 5% BSA; A300-846A; Bethyl Laboratories, Montgomery, USA) or anti-β-actin (1:10,000 in 5% BSA; A5441; Sigma), and then with either anti-goat-HRP (horse radish peroxidase; 1:10,000 in 5% milk; ab97110; Abcam, Cambridge, UK), anti-mouse-HRP (1:5,000 in 5% milk; 7076S; New England Biolabs) or anti-rabbit-HRP (1:5,000 in 5% milk; 7074S; New England Biolabs). The EZ-ECL Chemiluminescence Detection Kit for HRP (Gene Flow, Lichfield, UK) was used for protein detection on the Fujifilm LAS-4000 Luminescent Image Analyser (GE Healthcare).

**Cell viability assays.** KMS11 transduced with shRNA-CDCA7L-1, shRNA-CDCA7L-2 or shRNA-NSC were plated at 30,000 cells ml$^{-1}$. shRNA expression was induced with the addition of 1 μg ml$^{-1}$ doxycycline at $t = 0$ h. Cell viability was measured every 24 h by Trypan Blue exclusion and cytometry. In transduced KMS11 cells apoptosis was estimated using Annexin V/DAPI double staining assay Kit I (BD Biosciences, San Jose, USA) after 72 h induction following the manufacturer's recommendations. Samples were analysed using a BD LSRII flow cytometer (BD Bioscience).

**Cell cycle analysis.** After 72 h doxycycline induction-transduced KMS11 cells were fixed in 1 ml ice-cold 70% ethanol for 15 min, and then washed and resuspended in 860 μl PBS, 100 μl RNase (1 mg ml$^{-1}$; Sigma) and 40 μl PI (1 mg ml$^{-1}$; Sigma). After incubation at 37 °C for 25 min, cells were analysed on a BS LSRII flow cytometer (BD Bioscience).

**Association between *CDCA7L* expression and patient outcome.** To examine the relation between *CDCA7L* expression and patient outcome, we made use of data from five independent patient cohorts GSE9782 ($n = 265$), GSE2658 ($n = 559$), GSE19784 ($n = 320$), E-MTAB-372 ($n = 246$) and GSE21349 ($n = 183$). For each data set, the patients were grouped by their *CDCA7L* expression (upper and lower quartiles). Analysis was performed using the log-rank test to estimate expression-associated hazard ratio and the 95% confidence interval. Overall tatistical significance tests were performed by combining the results for each data set under a fixed-effects model. Statistical test was conducted on R software version 3.1.3.

**Data availability.** Existing patient expression data that support the findings of this study are deposited in GEO with accession codes GSE21349, GSE2658, GSE31161, GSE9782, GSE19784, and in EMBL-EBI with accession code E-MTAB-2299, E-MTAB-372.

ChIP-seq data that support the findings of this study are deposited in GEO with accession codes: GSE36354, GSE53215, GSE32465 and GSE29611.

Epigenomic profiling data that support the findings of this study are deposited in GEO with accession codes: DNaseI HS and FAIRE: GSE40833,ChromHMM: GSE38163.

Sequence data that support the findings of this study are deposited in EGA with accession code: EGAS00001001147. The remaining data are available within the Article and Supplementary Information files or available from the authors upon request.

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

## Acknowledgements

Myeloma UK provided principal funding. Additional funding was provided by Bloodwise, Cancer Research UK (C1298/A8362 supported by the Bobby Moore Fund), The Rosetrees Trust, Dietmar Hopp Foundation and the German Ministry of Education and Science (BMBF: CLIOMMICS (01ZX1309).

## Author contributions

N.L., D.C.J. and R.S.H. drafted the manuscript; N.L., D.C.J. and R.S.H. designed the study; R.S.H. and G.J.M. obtained funding; N.L., J.B.S., G.O. and F.M. performed laboratory work; H.G., G.J.M., K.H. and M.K. performed sample ascertainment and provided data; N.L., D.C.J., J.S.M., T.M. and N.W. performed bioinformatics and statistical analyses. All authors contributed to the final manuscript.

## Additional information

**Competing financial interests:** The authors declare no competing financial interests.

