## [Peer Review File · Nature Communications]

Reviewers' comments:

Reviewer #1 (Remarks to the Author): GWAS expert

This report provides functional characterization of SNP rs4487645 at 7p15.3, which has been seen to be the strongest GWAS association with MM risk. Using a comprehensive set of analyses and experiments, they show evidence for the 7p15.3 association and MM through an extensive pathway involving CDCA7L, IRF4 and MYC. In particular, they describe a novel functional interaction between CDCA7L and IRF4 in MM. The design of the studies are generally appropriate and conclusions follow logically from the results of their experiments. There are only minor concerns regarding clarification and interpretation of results. These are listed below:

1) Several analyses are performed on the same datasets/trials, and it would be helpful to include a supplemental table that details the different studies, sample size, years of ascertainment, types of MM cases (i.e. incident, etc), available germline and somatic data (RNA-Seq of CD138+ cells, GWAS, etc.) and provides references to these studies. This table would also help with interpretation of differences seen across studies (i.e. Suppl Figs 2, 3, 8).

2) Please include the frequency of the rs4487645 risk allele and appreciation for the extent of MM cases that may be influenced by this pathway.

3) Page 3, line 65. Please clarify source of mRNA expression (from CD138+ plasma cells).

4) Is there anything unique about the cell lines (i.e. KMS11) used for a number of the assays here? Are results expected to be similar in other selected MM cell lines?

5) Page. 5 and 14. The analyses of CDCA7L expression and MM survival should be further described (see #1 above). In particular, were these newly diagnosed cases and pre-treatment expression; any adjustment for age or other factors; what were the number of events; are the age and genders similar, etc.?

6) Page 9, Line 242-250. It is not clear why only the extreme quartiles were used to assess the relationship between IRF4 and CDCA7L gene expression across the five datasets and not all the data (i.e. correlations which would provide even greater evidence).

Reviewer #2 (Remarks to the Author): Expert in functional genomics

Summary of the key results: This study demonstrates how the GWAS signal for multiple myeloma (MM) found on 7p15.3 is most likely to be derived from the SNP rs4487645, down to expression, genetic, epigenetic and bioinformatic data. It indicates how the target gene is likely to be CDCA7L, through expression and physical, 3C analysis. It also demonstrates how this SNP differentially binds IRF4, how suppression of Myc down regulates the target gene CDCA7L and how suppression of CDCA7L itself leads to cell apoptosis and an adverse patient outcome.

Originality and interest. This work is of great interest to researchers both in the 'post-GWAS' era of attempting to link association signals with causal SNPs, genes, mechanisms and pathways. It is also of interest to researchers in the field of MM, showing, as it does a likely role for IRF4/MyC pathway in CDCA7L driven disease.

Data & methodology: All the data and methodology is sound and appropriate.

Appropriate use of statistics. All statistics are appropriate

Conclusions. Conclusions are appropriate and not over-stated

Suggested improvements. Minor improvement - Supplementary figure 2 is referenced after figure 3 in the text. It is not clear, to me at least, exactly where the clinical data is from (GSE9782). Could this be clearly indicated/referenced. Finally, I think probably spuriously, the SNP rs4487645, is labelled as not being bi-allelic on some data-bases (e.g. dbSNP/ LDlink). Could the authors clarify that it is indeed a bi-allelic SNP (G>T)?

References. ok

Clarity and context. I found the manuscript very clear, easy to read and follow.

Reviewer #3 (Remarks to the Author): Expert in functional genomics and TF occupancy

The authors proposed a mechanism on how a SNP, rs4487645, in 7p15.3 may regulate a gene, CDCA7L. The authors demonstrated that the risk allele of the SNP confers stronger binding of IRF4, which upregulated the expression of CDCA7L. CDCA7L interacts with MYC and the authors proposed that there is a complex network between IRF4, CDCA7L and MYC, and this interaction may be modulated by rs4487645.

Major:

Did the authors examine the effect of rs4487645 on DNase11? What other genes this enhancer region might regulate? This is possible to predict using bioinformatics approaches (cis-eQTL, promoter-enhancer interaction etc.).

How strong is the binding of IRF4 at the enhancer? If the binding motif for PU1 is available, can the authors show the effect of rs4487645 on PU1 binding motif? Please show the position of the SNP in the binding of IRF4 using IRF4 ChIP-seq data (add the IRF4 track in figure 1?).

Authors may consider performing allele-specific 3C to validate that the looping between rs4487645 enhancer and CDCA7L promoter is indeed dependent on the risk allele of the SNP.

Author mentioned the association p-value for rs4487645 with MM is 5.30×10^{-25} . The p-value is in the magnitude of 10^{-9} and 10^{-14} in two different studies as listed in GWAS catalog.

Minor:

The x-axis label in Figure 3 – should that be "Position" instead of "Distance"? If Position, should this be "Mb" instead of "Kb"?

Please elaborate on the imputation method as mentioned in the last paragraph of page 2.

Please elaborate on the method for performing the association testing conditional on rs4487645 as mentioned in the last line of page 2.

Please add the level of significance for fig 4a.

Supplementary figure 2 is referred after Supplementary figure 3 and 4 in the main text.

Response to Referees

Re: “Multiple myeloma risk variant at 7p15.3 creates IRF4-binding site and interferes with c-Myc-interacting CDCA7L expression”

We are grateful to the three reviewers for their comments and have made the below changes to our manuscript.

Reviewer #1

This report provides functional characterization of SNP rs4487645 at 7p15.3, which has been seen to be the strongest GWAS association with MM risk. Using a comprehensive set of analyses and experiments, they show evidence for the 7p15.3 association and MM through an extensive pathway involving CDCA7L, IRF4 and MYC. In particular, they describe a novel functional interaction between CDCA7L and IRF4 in MM. The design of the studies are generally appropriate and conclusions follow logically from the results of their experiments. There are only minor concerns regarding clarification and interpretation of results. These are listed below:

Response: We appreciate that the reviewer found our paper of interest.

1.1. Several analyses are performed on the same datasets/trials, and it would be helpful to include a supplemental table that details the different studies, sample size, years of ascertainment, types of MM cases (i.e. incident, etc), available germline and somatic data (RNA-Seq of CD138+ cells, GWAS, etc.) and provides references to these studies. This table would also help with interpretation of differences seen across studies (i.e. Suppl Figs 2, 3, 8).

Response: We have now provided this information in a new Supplementary Table 3.

1.2. Please include the frequency of the rs4487645 risk allele and appreciation for the extent of MM cases that may be influenced by this pathway.

Response: As requested this information has been added to the main text. Specifically, the risk allele frequency and odds ratio.

1.3. Page 3, line 65. Please clarify source of mRNA expression (from CD138+ plasma cells).

Response: As requested this information has been added to the main text.

1.4. Is there anything unique about the cell lines (i.e. KMS11) used for a number of the assays here? Are results expected to be similar in other selected MM cell lines?

Response: KMS11 and GM11992 are heterozygous for SNP rs4487645. There is no subtype-specific association (karyotype) for rs4487645 (referenced in Mitchell et al., Nature Communications, 2016). Hence we would assert that these myeloma cell lines are suitable for the analyses performed herein.

1.5. Page. 5 and 14. The analyses of CDCA7L expression and MM survival should be further described (see #1 above). In particular, were these newly diagnosed cases and pre-treatment expression; any adjustment for age or other factors; what were the number of events; are the age and genders similar, etc.?

Response: GSE2658, GSE9782, GSE19784, and E-MTAB-372 comprise of transplant eligible age (younger and fitter) patients, these sets have been comprehensively examined for use in survival expression signatures by Chng et al. (Leukemia, 2016) and Meißner et al. (Clinical Cancer Research, 2011). These studies have previously shown that these sets are comparable and can be used correlate biological related survival.

1.6. Page 9, Line 242-250. It is not clear why only the extreme quartiles were used to assess the relationship between IRF4 and CDCA7L gene expression across the five datasets and not all the data (i.e. correlations which would provide even greater evidence).

Response: We sought to highlight the expression correlation of *CDCA7L* and *IRF4* by analysing the upper and lower quartiles. Such analyses have been reported by others (Shalapour et al., Haematologica, 2011; Fujikawa et al., Blood, 2016). For the reviewers information Pearson correlation between gene expression generates similar results. Hence we have not been disingenuous.

CDCA7L & IRF4			
	r^2	p	p_boxplot
GSE21349	-0.093	0.21	0.56
E-MTAB-372	0.135	0.034	0.0096
GSE9782	0.13	0.039	2.69E-04
GSE2658	-0.023	0.59	0.67
GSE19784	0.012	0.82	0.51
CDCA7L & MYC			
	r^2	p	p_boxplot
GSE21349	0.04	0.56	0.29
E-MTAB-372	0.29	3.51E-06	0.0032
GSE9782	0.15	0.012	0.097
GSE2658	0.22	1.60E-07	3.10E-04
GSE19784	0.043	0.44	0.95

Reviewer #2

Summary of the key results: This study demonstrates how the GWAS signal for multiple myeloma (MM) found on 7p15.3 is most likely to derived from the SNP rs4487645, down to expression, genetic, epigenic and bioinformatic data. It indicates how the target gene is likely to be CDCA7L, through expresion and physical, 3C analysis. It also demonstrates how this SNP differentially binds IRF4, how suppression of Myc down regulates the target gene CDCA7L and how suppression of CDCA7L itself leads to cell apoptosis and an adverse patient outcome.

Originality and interest. This work is of great interest to researchers both in the 'post-GWAS' era of attempting to link association signals with casual SNPs, genes, mechanisms and pathways. It is also of interest to researchers in the field of MM, showing, as it does a likely role for IRF4/MyC pathway in CDCA7L driven disease.

Data & methodology: All the data and methodology is sound and appropriate.

Appropriate use of statistics. All statistics are appropriate

Conclusions. Conclusions are appropriate and not over-stated

Suggested improvements.

Minor improvement –

2.1 Supplementary figure 2 is referenced after figure 3 in the text.

Response: We apologise for this error which has been corrected.

2.2 It is not clear, to me at least, exactly where the clinical data is from (GSE9782). Could this be clearly indicated/referenced.

Response: We have now added this information as a new Supplementary Table 3.

2.3 Finally, I think probably spuriously, the SNP rs4487645, is labelled as not being bi-allelic on some data-bases (e.g. dbSNP/ LDlink). Could the authors clarify that it is indeed a bi-allelic SNP (G>T)?

Response: In HapMap CEU population this SNP is biallelic, frequency of the A-allele of rs4487645 (G>T) is 0.0 (dbSNP).

References. ok

Clarity and context. I found the manuscript very clear, easy to read and follow.

Reviewer #3

The authors proposed a mechanism on how a SNP, rs4487645, in 7p15.3 may regulate a gene, CDCA7L. The authors demonstrated that the risk allele of the SNP confers stronger binding of IRF4, which upregulated the expression of CDCA7L. CDCA7L interacts with MYC and the authors proposed that there is a complex network between IRF4, CDCA7L and MYC, and this interaction may be modulated by rs4487645.

Major:

3.1 Did the authors examine the effect of rs4487645 on DNAH11? What other genes this enhancer region might regulate? This is possible to predict using bioinformatics approaches (cis-eQTL, promoter-enhancer interaction etc.).

Response: We acknowledge this point however we have previously shown no eQTL for *DNAH11*. We now provide eQTL data for all genes within a 1Mb region flanking rs4487645 which shows that only *CDCA7L* displays a significant association (Supplementary Table 1).

3.2 How strong is the binding of IRF4 at the enhancer? If the binding motif for PU1 is available, can the authors show the effect of rs4487645 on PU1 binding motif? Please show the position of the SNP in the binding of IRF4 using IRF4 ChIP-seq data (add the IRF4 track in figure 1?).

Response: The strength of IRF4 binding at the enhancer is compared against background binding with isotype IgG IP control and intergenic region control in Figure 2b. The IRF4/PU.1-binding motif ETS/ISRE-consensus element, where rs4487645 is situated, is a fused binding motif recognised by both IRF4 and PU.1 equally as predicted by USCS Genome Browser. IRF4 ChIP-seq track has now been added to Figure 1 as requested.

3.3 Authors may consider performing allele-specific 3C to validate that the looping between rs4487645 enhancer and *CDCA7L* promoter is indeed dependent on the risk allele of the SNP.

Response: We would assert that allele-specific 3C in this context does not change the conclusion of the paper.

3.4 Author mentioned the association p-value for rs4487645 with MM is 5.30×10^{-25} . The p-value is in the magnitude of 10^{-9} and 10^{-14} in two different studies as listed in GWAS catalog.

Response: These *P*-values are those we previously reported from smaller studies Broderick et al. (Nature Genetics, 2011). These are referenced but have been subsumed into our recent large scale meta-analysis which we also reference (Mitchell et al., Nature Communications, 2016).

Minor:

3.5 The x-axis label in Figure 3 – should that be “Position” instead of “Distance”? If Position, should this be “Mb” instead of “Kb”?

Response: This error has been corrected.

3.6 Please elaborate on the imputation method as mentioned in the last paragraph of page 2.

Response: Comprehensive details are provided in Mitchell et al. (Nature communications, 2016) which we reference.

3.7 Please elaborate on the method for performing the association testing conditional on rs4487645 as mention in the last line of page 2.

Response: Comprehensive details are provided in Mitchell et al. (Nature communications, 2016) which we reference.

3.8 Please add the level of significance for fig 4a.

Response: Figure 4a shows growth curves. As stated in the Figure legend the significance at the 72 hour time-point is detailed Figure 4b.

3.9 Supplementary figure 2 is referred after Supplementary figure 3 and 4 in the main text.

Response: This error has been corrected.

REVIEWERS' COMMENTS:

Reviewer #1 (Remarks to the Author):

The authors have adequately addressed the majority of reviewer comments.

Small additions are requested for Supplemental Table 3 which describes the study characteristics including age (mean/range), gender (%) and race/ethnicity.

Reviewer #3 (Remarks to the Author):

This reviewer has no further comments.